# Adverse events of special interest and mortality following vaccination with mRNA (BNT162b2) and inactivated (CoronaVac) SARS-CoV-2 vaccines in Hong Kong: A retrospective study

**Carlos King Ho Wong**[1,2,3☯], **Kristy Tsz Kwan Lau**[1☯], **Xi Xiong**[1], **Ivan Chi Ho Au**[1], **Francisco Tsz Tsun Lai**[1,3], **Eric Yuk Fai Wan**[1,2,3], **Celine Sze Ling Chui**[3,4,5], **Xue Li**[1,3,6], **Esther Wai Yin Chan**[1,3], **Le Gao**[1], **Franco Wing Tak Cheng**[1], **Sydney Chi Wai Tang**[6], **Ian Chi Kei Wong**[1,3,7]*

**1** Centre for Safe Medication Practice and Research, Department of Pharmacology and Pharmacy, Li Ka Shing Faculty of Medicine, University of Hong Kong, Hong Kong SAR, China, **2** Department of Family Medicine and Primary Care, School of Clinical Medicine, Li Ka Shing Faculty of Medicine, University of Hong Kong, Hong Kong SAR, China, **3** Laboratory of Data Discovery for Health, Hong Kong Science and Technology Parks, Hong Kong SAR, China, **4** School of Nursing, Li Ka Shing Faculty of Medicine, University of Hong Kong, Hong Kong SAR, China, **5** School of Public Health, Li Ka Shing Faculty of Medicine, University of Hong Kong, Hong Kong SAR, China, **6** Department of Medicine, School of Clinical Medicine, Li Ka Shing Faculty of Medicine, University of Hong Kong, Hong Kong SAR, China, **7** Centre for Medicines Optimisation Research and Education, Research Department of Practice and Policy, UCL School of Pharmacy, University College London, London, United Kingdom

☯ These authors contributed equally to this work.
* wongick@hku.hk

**Data Availability Statement:** The data used in this study are not freely available. Approvals for the use of data were obtained from the Department of

## Abstract

### Background

Safety monitoring of coronavirus disease 2019 (COVID-19) vaccines is crucial during mass vaccination rollout to inform the choice of vaccines and reduce vaccine hesitancy. Considering the scant evidence directly comparing the safety profiles of mRNA and inactivated SARS-CoV-2 vaccines, this territory-wide cohort study aims to compare the incidence of various adverse events of special interest (AESIs) and all-cause mortality between Corona-Vac (inactivated vaccine) and BNT162b2 (mRNA-based vaccine). Our results can help vaccine recipients make an informed choice.

### Methods and findings

A retrospective, population-based cohort of individuals who had received at least 1 dose of BNT162b2 or CoronaVac from 23 February to 9 September 2021 in Hong Kong, and had data linkage to the electronic medical records of the Hong Kong Hospital Authority, were included. Those who had received mixed doses were excluded. Individuals were observed from the date of vaccination (first or second dose) until mortality, second dose vaccination (for first dose analysis), 21 days after vaccination, or 30 September 2021, whichever came

Health and the Hospital Authority specifically for this COVID-19 vaccine safety monitoring study. Authors are bound by ethical, legal and contractual conditions imposed by both Department of Health and the Hospital Authority, and are not allowed to use the data for any other purposes or divulge the data to any third parties. The vaccination record data are owned by the Department of Health. Clinical records are owned by Hospital Authority. Vaccination records were linked to clinical records on de-identified patients of the Hospital Authority. Following approvals from the Institutional Review Board, data requests were submitted and assessed by both Department of Health and Hospital Authority prior to data release for use by specified research delegates only. For further information regarding the data request and approval process, please see: (https://www3.ha.org.hk/data/Provision/Submission). Hospital Authority data access inquiries can be directed to hacpaaedr@ha.org.hk.

**Funding:** This study was funded by a research grant from the Food and Health Bureau, The Government of the Hong Kong Special Administrative Region (ICKW, Ref. No. COVID19F01). The funders had no role in study design, data collection and analysis, decision to publish, or preparation of the manuscript.

**Competing interests:** I have read the journal's policy and the authors of this manuscript have the following competing interests: CKHW reports receipt of research funding from the EuroQoL Group Research Foundation, the Hong Kong Research Grants Council, and the Hong Kong Health and Medical Research Fund; FTTL has been supported by the RGC Postdoctoral Fellowship under the Hong Kong Research Grants Council; EYFW has received research grants from the Food and Health Bureau of the Government of the Hong Kong SAR, and the Hong Kong Research Grants Council, outside the submitted work; CSLC has received grants from the Food and Health Bureau of the Hong Kong Government, Hong Kong Research Grant Council, Hong Kong Innovation and Technology Commission, Pfizer, IQVIA, and Amgen, personal fee from Primevigilance Ltd., outside the submitted work; XL has received research grants from the Food and Health Bureau of the Government of the Hong Kong SAR, research and educational grants from Janssen and Pfizer, internal funding from University of Hong Kong, consultancy fee from Merck Sharp & Dohme, unrelated to this work; EWYC reports grants from Research Grants Council (RGC, Hong Kong), grants from Research Fund Secretariat of the Food and Health Bureau, grants from National

first. Baseline characteristics of vaccinated individuals were balanced between groups using propensity score weighting. Outcome events were AESIs and all-cause mortality recorded during 21 days of post-vaccination follow-up after each dose, except anaphylaxis, for which the observation period was restricted to 2 days after each dose. Incidence rate ratios (IRRs) of AESIs and mortality comparing between CoronaVac and BNT162b2 recipients were estimated after each dose using Poisson regression models. Among 2,333,379 vaccinated individuals aged 18 years or above, the first dose analysis included 1,308,820 BNT162b2 and 955,859 CoronaVac recipients, while the second dose analysis included 1,116,677 and 821,560 individuals, respectively. The most frequently reported AESI among CoronaVac and BNT162b2 recipients was thromboembolism (first dose: 431 and 290 per 100,000 person-years; second dose: 385 and 266 per 100,000 person-years). After the first dose, incidence rates of overall AESIs (IRR = 0.98, 95% CI 0.89–1.08, $p = 0.703$) and mortality (IRR = 0.96, 95% CI 0.63–1.48, $p = 0.868$) associated with CoronaVac were generally comparable to those for BNT162b2, except for Bell palsy (IRR = 1.95, 95% CI 1.12–3.41, $p = 0.018$), anaphylaxis (IRR = 0.34, 95% CI 0.14–0.79, $p = 0.012$), and sleeping disturbance or disorder (IRR = 0.66, 95% CI 0.49–0.89, $p = 0.006$). After the second dose, incidence rates of overall AESIs (IRR = 0.97, 95% CI 0.87–1.08, $p = 0.545$) and mortality (IRR = 0.85, 95% CI 0.51–1.40, $p = 0.516$) were comparable between CoronaVac and BNT162b2 recipients, with no significant differences observed for specific AESIs. The main limitations of this study include residual confounding due to its observational nature, and the possibility of its being underpowered for some AESIs with very low observed incidences.

## Conclusions

In this study, we observed that the incidences of AESIs (cumulative incidence rate of 0.06%–0.09%) and mortality following the first and second doses of CoronaVac and BNT162b2 vaccination were very low. The safety profiles of the vaccines were generally comparable, except for a significantly higher incidence rate of Bell palsy, but lower incidence rates of anaphylaxis and sleeping disturbance or disorder, following first dose CoronaVac versus BNT162b2 vaccination. Our results could help inform the choice of inactivated COVID-19 vaccines, mainly administered in low- and middle-income countries with large populations, in comparison to the safety of mRNA vaccines. Long-term surveillance on the safety profile of COVID-19 vaccines should continue.

## Author summary

### Why was this study done?

- Various SARS-CoV-2 vaccines have been developed for coronavirus disease 2019 (COVID-19) prevention to reduce the risks of severe disease and death; however, vaccine hesitancy due to fear of adverse reactions remains a barrier to mass uptake.

- Continuous post-marketing surveillance of vaccine safety is crucial to guide informed decision-making and promote vaccine uptake in the community.

Natural Science Fund of China, grants from
Wellcome Trust, grants from Bayer, grants from
Bristol-Myers Squibb, grants from Pfizer, grants
from Janssen, grants from Amgen, grants from
Takeda, grants from Narcotics Division of the
Security Bureau of HKSAR and honorarium from
Hospital Authority, outside the submitted work;
SCWT reports research funding outside the
submitted work from the Hong Kong RGC, and the
Hong Kong Health and Medical Research Fund,
and National Natural Science Fund of China; ICKW
reports research funding outside the submitted
work from Amgen, Bristol-Myers Squibb, Pfizer,
Janssen, Bayer, GSK, Novartis, the Hong Kong
RGC, and the Hong Kong Health and Medical
Research Fund, National Institute for Health
Research in England, European Commission,
National Health and Medical Research Council in
Australia, and also received speaker fees from
Janssen and Medice in the previous 3 years; KTKL,
XX, ICHA, LG, and FWTC report no disclosures
relevant to the manuscript. Authors who report
research funding or grants from Pfizer outside the
submitted work have no financial conflict of
interests to the current study.

**Abbreviations:** AESI, adverse event of special
interest; CAD, coronary artery disease; COVID-19,
coronavirus disease 2019; FDA, US Food and Drug
Administration; GBS, Guillain-Barré syndrome;
IRR, incidence rate ratio; MI, myocardial infarction;
PS, propensity score; TM, transverse myelitis;
WHO, World Health Organization.

- There is very limited evidence directly comparing the safety profiles of mRNA-based and inactivated SARS-CoV-2 vaccines.

## What did the researchers do and find?

- A territory-wide retrospective cohort study of individuals who had received at least 1 dose of BNT162b2 (mRNA-based vaccine, Comirnaty) or CoronaVac (inactivated SARS-CoV-2 vaccine) from 23 February to 9 September 2021 in Hong Kong was conducted to compare the occurrence of selected adverse events of special interest (AESIs) and all-cause death between recipients of the 2 vaccines during 21 days of follow-up after the first and second doses.

- The incidence of overall AESIs was very low for both vaccines (cumulative incidence rate of 0.06%–0.09%), and the most frequently reported AESI among CoronaVac and BNT162b2 recipients was thromboembolism (first dose: 431 and 290 per 100,000 person-years; second dose: 385 and 266 per 100,000 person-years).

- The incidence rates of overall AESIs and all-cause death were comparable between CoronaVac and BNT162b2 recipients after the first and second doses, except for a higher incidence rate of Bell palsy, and lower incidence rates of anaphylaxis and sleeping disturbance or disorder, following first dose CoronaVac versus BNT162b2 vaccination.

## What do these findings mean?

- Both vaccines had similarly acceptable safety profiles, and the risks of AESIs and all-cause death following CoronaVac or BNT162b2 vaccination were very low, which adds to the real-world evidence on the safety of both COVID-19 vaccines, and may help reduce vaccine hesitancy by addressing safety concerns.

- Our results may inform the public about the choice of COVID-19 vaccines, and governments about the allocation of healthcare resources for monitoring specific AESIs during vaccine rollout (especially for Bell palsy among first dose CoronaVac recipients, and anaphylaxis and sleeping disturbance or disorder among first dose BNT162b2 recipients).

- As currently done for mRNA vaccines, clinical data and observational reports of all COVID-19 vaccines should be made publicly available in a timely manner, and long-term monitoring of their safety profiles should continue.

## Introduction

Since the outbreak of the coronavirus disease 2019 (COVID-19) pandemic, an unprecedented number of SARS-CoV-2 vaccines using different platforms and with varying efficacy have been developed, namely inactivated virus, mRNA, viral vector, and protein subunit vaccines [1]. While rapid and mass vaccination of individuals is essential to reducing COVID-

19-related hospitalizations, disease severity, and associated deaths, vaccine hesitancy has been observed owing to fear of adverse reactions following immunization [2]. Accordingly, continuous monitoring of vaccine safety via active and passive surveillance is crucial to providing up-to-date evidence about any potential safety signals over the rollout period, especially those concerning adverse events of special interest (AESIs).

In Hong Kong SAR (Special Administrative Region), China, the government initiated a territory-wide vaccination program with CoronaVac (inactivated SARS-CoV-2 vaccine) on 23 February 2021, and with BNT162b2 (mRNA-based vaccine, equivalent to Comirnaty) on 6 March 2021; individuals can choose to get vaccinated with either platform. Adopting the traditional approach, CoronaVac was developed as an inactivated whole-virion SARS-CoV-2 vaccine using the adjuvant aluminum hydroxide, and its tolerability has been demonstrated in several clinical trials [3–5]. Two cross-sectional studies conducted among healthcare workers in Turkey and China reported that the prevalence of local and systemic side effects for CoronaVac was significantly lower than for mRNA vaccines, and all adverse effects, including allergic reaction and lymphadenopathy, were mild and transient [6,7]. Upon the rollout of CoronaVac in multiple countries, case reports of AESIs have emerged in the literature, for example, thyroiditis, Kounis syndrome, and kidney diseases [8–11]. Nevertheless, systematic evaluation of the incidence of various AESIs possibly associated with this inactivated vaccine—compared to other platforms, unvaccinated individuals, or background incidence rates in the population—is profoundly lacking.

BNT162b2 vaccine was the first SARS-CoV-2 vaccine approved by the US Food and Drug Administration (FDA) for COVID-19 prevention in adults [12]. Despite the novel mRNA platform, substantial evidence on its safety and effectiveness is accumulating, given the rapid deployment and massive number of doses administered across different populations worldwide. While vaccine reactogenicity was transient, with mainly mild to moderate adverse effects, serious adverse events such as lymphadenopathy, arrhythmia, and leg paresthesia were recorded among BNT162b2 recipients in the clinical trial [13]. Also, strong pharmacoepidemiological data have suggested an excess risk of myocarditis and pericarditis [14–16], lymphadenopathy, herpes zoster infection, and potentially appendicitis with this mRNA vaccine [14]. Concurrently with mass vaccination, several AESIs in close temporal relationship with BNT162b2 have been identified for further detection of safety signals, including anaphylaxis [15,17], myocarditis and pericarditis [18], kidney diseases [19,20], Bell palsy [14,21,22], Guillain-Barré syndrome (GBS) [17], seizure [15,17], transverse myelitis (TM) [15,17], venous thromboembolism, thrombosis, and thrombocytopenia [15,23].

Inactivated and mRNA vaccines are both associated with risks of serious adverse events comparable to placebo control [24–27], yet there is a lack of direct comparison of the safety profiles of these 2 vaccine platforms widely used around the world. A nested case–control study conducted in Hong Kong has suggested an elevated risk of Bell palsy following CoronaVac vaccination, which was not evident with BNT162b2 vaccination [28]. This is in line with a disproportionality analysis that used the World Health Organization (WHO) pharmacovigilance database and concluded that a signal of facial paralysis had not been displayed for mRNA COVID-19 vaccines compared to other viral vaccines [29]. While the WHO pharmacovigilance database may be used for generating safety signals for further testing and confirmation, it relies on the spontaneous reporting of suspected adverse drug reactions by patients and health professionals of participating countries, which has been extensively performed for BNT162b2 but not as much for CoronaVac [30]. In view of the very limited evidence comparing the safety profile of mRNA and inactivated SARS-CoV-2 vaccines, our population-based, retrospective cohort study aims to evaluate and compare the incidences of AESIs following the first and second doses of each vaccine type, with reference to the lists recommended for

continuous monitoring of COVID-19 vaccine safety [31]. The findings of this study could inform the public in decision-making on the choice of COVID-19 vaccines, reduce vaccine hesitancy by addressing safety concerns, and inform governments about the allocation of healthcare resources for AESI surveillance and monitoring.

## Methods

### Study design and study population

A territory-wide cohort study was conducted to compare the incidence rates of AESIs and all-cause mortality between CoronaVac (supplied by Sinovac) and BNT162b2 (supplied by Fosun Pharma as Comirnaty vaccines, manufactured by BioNTech in Germany) [32] vaccination after the first and second doses.

The mass COVID-19 vaccination program in Hong Kong was launched on 23 February 2021 for CoronaVac and 6 March 2021 for BNT162b2. Certain patient groups were prioritized to receive vaccination at the start and during specific periods of the community vaccination program. However, there were no guidelines recommending that certain individuals or patient groups receive a particular vaccine or different service delivery. Individuals remained free to choose their preferred type of vaccine at their first dose, but were unable to switch vaccine type for their second dose (except for rare instances of anaphylaxis following the first dose, where switching between vaccine types is allowed on a case-by-case basis). People aged 18 years or above who had received at least 1 dose of CoronaVac or BNT162b2 vaccine in Hong Kong SAR, China, between 23 February and 9 September 2021 were included in the study (Fig 1). People who had received BNT162b2 as the first dose followed by CoronaVac as the second dose (or vice versa) were excluded from the second dose analysis.

### Data sources

Anonymized, population-wide COVID-19 vaccination records obtained from the Department of Health and electronic medical records retrieved from the Hong Kong Hospital Authority (the statutory body managing public healthcare services in the region) were linked using a unique mapping key (Hong Kong Identity Card number or foreign passport number). Vaccination records included the brand of vaccine, venue for vaccination, and date of administration. Centralized electronic medical records included patient demographics, date of registered death (including both hospital and non-hospital deaths), drug dispensing records, diagnoses, procedures, and laboratory tests. The validity of disease diagnoses identified from electronic medical records has been shown to be high [33–38]. These 2 linked sources of data have been extensively used for pharmacovigilance and COVID-19 vaccine safety studies [18,28,39–45].

### Ethics approval

Ethical approval for this study was granted by the Institutional Review Board of the University of Hong Kong/Hospital Authority Hong Kong West Cluster (UW 21–149 and UW 21–138) and the Department of Health Ethics Committee (LM 21/2021). A waiver of participant consent was granted for this retrospective cohort study using anonymized data.

### Exposure period of the first and second doses

For individuals who received the first dose of BNT162b2 or CoronaVac from 23 February to 9 September 2021, the follow-up period was from the date of first dose vaccination (baseline date) until mortality, second dose vaccination, 21 days after the first dose, or the data cutoff

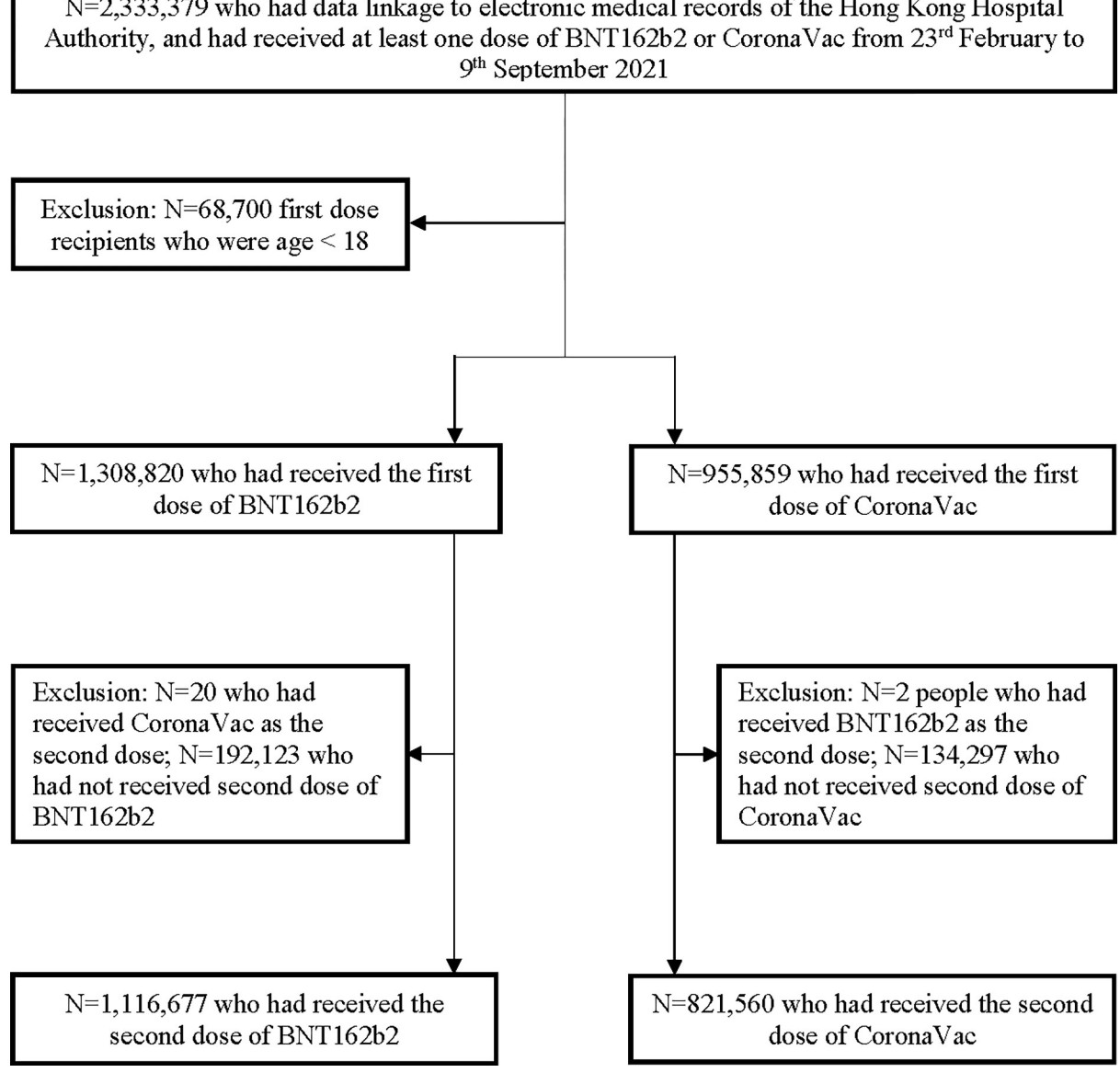

**Fig 1. Inclusion and exclusion criteria for the analysis of individuals who received a first or second dose of BNT162b2 or CoronaVac from 23 February to 9 September 2021 in Hong Kong SAR, China.**

date (30 September 2021), whichever came first. For those who received the second dose of either vaccine within the inclusion period, their observation period was stratified into 2 exposure risk periods, namely follow-up windows for the first and second doses. The follow-up period for those who received a second dose was from the date of second dose vaccination (baseline date) until mortality, 21 days after the second dose, or the data cutoff date, which-ever came first. Vaccinated individuals had at most 21 days of post-vaccination follow-up after each dose, to make the follow-up periods comparable for the first and second doses and between vaccines, since approximately two-thirds of BNT162b2 recipients had their second dose administered 21 days after their first dose, which is the recommended dosing interval for this mRNA vaccine [46].

## Outcomes

Our study outcomes adapted the full list of AESI outcomes endorsed by the WHO Global Advisory Committee on Vaccine Safety, which has been widely used for the safety surveillance of COVID-19 vaccines; a range of events suggested by the European Medicines Agency–funded vACCine covid-19 monitoring readinESS (ACCESS) project; and "subacute thyroiditis," "pancreatitis," and "rhabdomyolysis," as updated by the Coalition for Epidemic Preparedness Innovations–funded Safety Platform for Emergency vACcines (SPEAC) project [31,47,48]. These last 3 AESI outcomes have not yet been endorsed by the WHO but were included in this study following discussion and consensus reached between co-investigators and the Department of Health. All of the AESI outcomes have been identified as events potentially related to existing marketed vaccines, events related to vaccine platforms or adjuvants, or events that may be associated with COVID-19.

Our study outcomes consisted of various AESIs including autoimmune diseases, cardiovascular system diseases, circulatory system diseases, hepato-renal system diseases, nerve and central nervous system diseases, and respiratory system diseases. The list of AESIs investigated in this study is shown in S1 Table, where cases are defined by International Classification of Diseases–9th Revision, Clinical Modification (ICD-9-CM) diagnosis and procedure codes and laboratory parameters. Previous research validating ICD-9-CM diagnostic codes in Hospital Authority data estimated the positive predictive value at >85% [33]. The Hospital Authority database has previously been used extensively for COVID-19 vaccination safety and pharmacovigilance studies, one of which quantified risks of AESIs following COVID-19 vaccination [39,40]. Outcome events were recorded during the 21 days of post-vaccination follow-up, except for anaphylaxis, for which the observation period was restricted to 2 days after each vaccine dose. The secondary outcome of interest was all-cause mortality.

## Covariates

Various demographic and clinical characteristics of the vaccinated individuals were considered in this analysis. As reported by previous studies [40,45,49–51], baseline covariates that were potential risk factors of AESIs were chosen. The following covariates were weighted between BNT162b2 and CoronaVac recipients for each dose (as elaborated the "Statistical analyses" section below): age, sex, any previous SARS-CoV-2 infection (defined as ever a positive result on the SARS-CoV-2 reverse transcription polymerase chain reaction [RT-PCR] test before COVID-19 vaccination), pre-existing comorbidities documented from 2018 (myocardial infarction [MI], peripheral vascular disease, cerebrovascular disease, coronary artery disease [CAD], chronic obstructive pulmonary disease, dementia, paralysis, diabetes with and without chronic complications, hypertension, chronic renal failure, mild and moderate-severe liver disease, ulcers, rheumatoid arthritis or other inflammatory polyarthropathy, acquired immune deficiency syndrome, malignancy, and metastatic solid tumor), medication use in the past 90 days (including renin–angiotensin-system agents, beta blockers, calcium channel blockers, diuretics, lipid-lowering agents, insulin, antidiabetic drugs, anticoagulants, antiplatelets, hormonal agents, antidepressants, non-steroidal anti-inflammatory drugs, drugs for gout, antiepileptic drugs, antiviral drugs, antibacterial drugs, and immunosuppressants), venue for vaccination (community vaccination center, clinic, or other), and the time interval between the administration of the first and second doses. Patients who had recovered from a previous SARS-CoV-2 infection were recommended to receive BNT162b2 at least 90 days after hospital discharge or CoronaVac at least 180 days after discharge [52]. There were no missing values in covariates, under the assumption that the absence of a medication record implies that the specific medication had neither been prescribed using the electronic prescribing system nor

dispensed by the Hospital Authority. Similar to other database studies, the absence of a disease diagnosis in an individual's lifetime records from the Hospital Authority database was treated as the individual being without that specific disease.

## Statistical analyses

This study is based on a regulatory pharmacovigilance program set up by the regulatory authority (Department of Health) to monitor the safety of COVID-19 vaccine emergency use. A prospective study protocol or analysis plan was not separately developed in the design of this study; however, this study was part of an overall research program that was assessed by the regulatory authority (Department of Health), clinical data custodian (Hospital Authority), institutional review board, and funding body (Food and Health Bureau). We developed a protocol (S1 Protocol) when designing this study, which was followed and used for independent double checking of data analysis to ensure data quality and reproducibility. Statistical analyses were planned in November 2021 and conducted in November 2021 and March 2022.

The variables considered in the propensity score (PS) model were potential risk factors for AESI after vaccination, including age, sex, Charlson Comorbidity Index, pre-existing comorbidities, history of medication use, and history of AESI. History of AESI was defined as a history of any of the conditions, except mortality, listed in S1 Table from 2005 to the date of first dose COVID-19 vaccination. The background rate of AESIs would influence both the vaccination rate and risk of AESI; hence, it was added to the PS model. The probability of receiving either CoronaVac or BNT162b2 was estimated by the PS model. Inverse probability of treatment weighting (IPTW) using the PS was applied, followed by truncation of the 1st and 99th percentiles of the observed PS weighting distribution to account for extreme weights [53] (S1 Fig). Standardized mean differences (SMDs) of baseline covariates between CoronaVac and BNT162b2 recipients were calculated, with SMD < 0.1 indicating covariate balance [54].

Inverse-probability-of-treatment-weighted Poisson regression models with person-years as an offset term were fitted to estimate the incidence rate ratios (IRRs) and corresponding 95% confidence intervals (CIs) of AESIs comparing between CoronaVac and BNT162b2 recipients after each vaccine dose. For patient confidentiality and robustness of the results, IRRs were estimated only when there were at least 5 events for a specific AESI in both the BNT162b2 and CoronaVac groups. A sensitivity analysis was performed on the study outcomes excluding people with previous SARS-CoV-2 infection, who might have had an increased risk of various disease diagnoses in association with SARS-CoV-2 infection. The goodness of fit of the Poisson regression models was assessed by deviance chi-squared tests and overdispersion tests, which were conducted during the revision following referees' comments [55].

All statistical analyses were performed using STATA version 16.0 (StataCorp, College Station, TX). A two-sided significance level of $p < 0.05$ was considered statistically significant.

This study is reported as per the Strengthening the Reporting of Observational Studies in Epidemiology (STROBE) guideline (S1 Checklist).

## Results

### Baseline characteristics

A total of 2,333,379 individuals had data linkage to electronic medical records of the Hong Kong Hospital Authority, and received at least 1 dose of BNT162b2 or CoronaVac from 23 February to 9 September 2021 (Fig 1). The first dose analysis included 1,308,820 BNT162b2 and 955,859 CoronaVac recipients. After excluding 22 recipients with mixed vaccine doses, 1,116,677 (85.3%) and 821,560 (85.9%) individuals were fully vaccinated with BNT162b2 and CoronaVac, respectively. BNT162b2 recipients were generally younger and healthier than

CoronaVac recipients (Table 1). The distribution of PSs in the 2 vaccine groups after each dose greatly overlapped after weighting (S2 Fig). All baseline characteristics were balanced between groups, with SMD < 0.1 after PS weighting. Among fully vaccinated individuals, the mean (standard deviation) dosing interval was 22.8 (4.3) and 29.2 (5.2) days for the BNT162b2 and CoronaVac groups, respectively.

## AESIs and all-cause mortality after the first dose

Overall AESI and mortality rates per 100,000 doses were 65 and 3 cases, respectively, for the first dose of BNT162b2 administered, and 90 and 6 cases for the first dose of CoronaVac administered. There were 861 (1,566 per 100,000 person-years) and 850 (1,162 per 100,000 person-years) AESI cases within 21 days after the first dose of CoronaVac and BNT162b2, respectively (Table 2). The cumulative incidence rate of overall AESIs was very low relative to the number of first vaccine doses administered, i.e., 0.06% for BNT162b2 and 0.09% for CoronaVac. Frequently reported AESIs among CoronaVac and BNT162b2 recipients after the first dose were thromboembolism (431 and 290 per 100,000 person-years), anaphylaxis (134 and 376 per 100,000 person-years), CAD (271 and 157 per 100,000 person-years), arrhythmia (236 and 168 per 100,000 person-years), MI (180 and 104 per 100,000 person-years), and sleeping disturbance or disorder (131 and 167 per 100,000 person-years).

AESIs and all-cause mortality reported after the first dose of CoronaVac were compared to BNT162b2 as a reference group (Table 3). Goodness-of-fit tests demonstrated adequate model fit for all Poisson regression models (S2 Table). Despite the slightly higher crude incidence rate of overall AESIs among first dose CoronaVac recipients than among BNT162b2 recipients, the incidence rates of overall (IRR = 0.98, 95% CI 0.89–1.08, $p$ = 0.703) and specific AESIs during 21 days post-vaccination were generally comparable for the first dose of CoronaVac and BNT162b2 after weighting, except for Bell palsy (67 versus 30 per 100,000 person-years; IRR = 1.95, 95% CI 1.12–3.41, $p$ = 0.018), anaphylaxis (IRR = 0.34, 95% CI 0.14–0.79, $p$ = 0.012), and sleeping disturbance or disorder (IRR = 0.66, 95% CI 0.49–0.89, $p$ = 0.006). Despite the slightly higher incidence rate of sleeping disturbance or disorder among BNT162b2 recipients than among CoronaVac recipients, the occurrence of this AESI was not concentrated in the days shortly after the first dose of BNT162b2 vaccination (S3 Fig). Hence its occurrence might not be correlated with initial reactogenicity. Meanwhile, no significant differences were identified for all-cause mortality comparing the first dose recipients of CoronaVac and BNT162b2 (IRR = 0.96, 95% CI 0.63–1.48, $p$ = 0.868).

## AESIs and all-cause mortality after the second dose

Overall AESI and mortality rates per 100,000 doses were, respectively, 66 and 3 cases for the second dose of BNT162b2 administered and 88 and 4 cases for the second dose of CoronaVac administered. There were 720 (1,524 per 100,000 person-years) and 733 (1,141 per 100,000 person-years) AESI cases within 21 days after the second dose of CoronaVac and BNT162b2, respectively (Table 4). Similar to the results for the first dose, the cumulative incidence rates of overall AESIs were very low relative to the number of second vaccine doses administered, i.e., 0.07% for BNT162b2 and 0.09% for CoronaVac. Again, despite the slightly higher crude incidence rate of overall AESIs among second dose CoronaVac versus BNT162b2 recipients, the incidence rates of overall AESIs (IRR = 0.97, 95% CI 0.87–1.08, $p$ = 0.545) and all-cause mortality (IRR = 0.85, 95% CI 0.51–1.40, $p$ = 0.516) were comparable between recipients of CoronaVac and BNT162b2 after weighting (Table 3). Frequently reported AESIs after second dose CoronaVac and BNT162b2 vaccination were thromboembolism (385 and 266 per 100,000 person-years), CAD (271 and 157 per 100,000 person-years), arrhythmia (220 and 157 per

**Table 1. Baseline characteristics of individuals who had received a first or second dose of BNT162b2 or CoronaVac from 23 February to 9 September 2021 in Hong Kong SAR, China.**

| Baseline characteristic | First dose recipients | | | | Second dose recipients | | | |
|---|---|---|---|---|---|---|---|---|
| | Before weighting | | | After weighting | Before weighting | | | After weighting |
| | Mean ± SD or N (%) | | SMD | SMD | Mean ± SD or N (%) | | SMD | SMD |
| | BNT162b2 (N = 1,308,820) | CoronaVac (N = 955,859) | | | BNT162b2 (N = 1,116,677) | CoronaVac (N = 821,560) | | |
| Age, years | 45.7 ± 16.0 | 55.3 ± 14.1 | 0.63 | 0.02 | 45.9 ± 15.7 | 54.8 ± 13.9 | 0.59 | 0.02 |
| Sex | | | 0.03 | 0.00 | | | 0.03 | 0.00 |
| Male | 584,158 (44.6%) | 439,928 (46.0%) | | | 502,740 (45.0%) | 383,164 (46.6%) | | |
| Female | 724,662 (55.4%) | 515,931 (54.0%) | | | 613,937 (55.0%) | 438,396 (53.4%) | | |
| Dosing interval, days | NA | NA | NA | NA | 22.8 ± 4.3 | 29.2 ± 5.2 | NA | NA |
| Venue for vaccination | | | NA | NA | | | NA | NA |
| Community vaccination center | 1,264,703 (98.5%) | 463,815 (50.3%) | | | 1,099,211 (98.4%) | 428,794 (52.2%) | | |
| Clinic | 0 (0.0%) | 441,526 (47.9%) | | | 0 (0.0%) | 378,865 (46.1%) | | |
| Other | 19,249 (1.5%) | 17,198 (1.9%) | | | 17,377 (1.6%) | 13,497 (1.6%) | | |
| Previous SARS-CoV-2 infection[a] | 4,642 (0.4%) | 1,690 (0.2%) | 0.03 | 0.00 | 422 (0.0%) | 263 (0.0%) | 0.00 | 0.00 |
| Pre-existing comorbidities | | | | | | | | |
| Charlson Comorbidity Index | 1.4 ± 1.5 | 2.2 ± 1.5 | 0.56 | 0.01 | 1.4 ± 1.5 | 2.1 ± 1.5 | 0.52 | 0.01 |
| History of AESI | 395,232 (30.2%) | 382,549 (40.0%) | 0.21 | 0.02 | 340,006 (30.4%) | 323,326 (39.4%) | 0.19 | 0.01 |
| Myocardial infarction | 1,662 (0.1%) | 2,238 (0.2%) | 0.03 | 0.00 | 1,347 (0.1%) | 1,688 (0.2%) | 0.02 | 0.00 |
| Peripheral vascular disease | 657 (0.1%) | 881 (0.1%) | 0.02 | 0.00 | 513 (0.0%) | 653 (0.1%) | 0.01 | 0.00 |
| Cerebrovascular disease | 11,591 (0.9%) | 17,638 (1.8%) | 0.08 | 0.00 | 9,374 (0.8%) | 13,259 (1.6%) | 0.07 | 0.00 |
| Coronary artery disease | 253,883 (19.4%) | 343,413 (35.9%) | 0.00 | 0.00 | 210,140 (18.8%) | 274,353 (33.4%) | 0.00 | 0.00 |
| Chronic obstructive pulmonary disease | 11,837 (0.9%) | 11,324 (1.2%) | 0.03 | 0.00 | 9,817 (0.9%) | 8,954 (1.1%) | 0.02 | 0.00 |
| Dementia | 251 (0.0%) | 607 (0.1%) | 0.02 | 0.00 | 203 (0.0%) | 261 (0.0%) | 0.01 | 0.00 |
| Paralysis | 321 (0.0%) | 525 (0.1%) | 0.02 | 0.00 | 252 (0.0%) | 359 (0.0%) | 0.01 | 0.00 |
| Diabetes without chronic complication | 75,440 (5.8%) | 98,194 (10.3%) | 0.17 | 0.00 | 62,113 (5.6%) | 77,702 (9.5%) | 0.15 | 0.00 |
| Diabetes with chronic complication | 2,366 (0.2%) | 3,135 (0.3%) | 0.03 | 0.00 | 1,832 (0.2%) | 2,350 (0.3%) | 0.03 | 0.00 |
| Hypertension | 177,913 (13.6%) | 225,107 (23.6%) | 0.08 | 0.00 | 148,978 (13.3%) | 182,379 (22.2%) | 0.07 | 0.00 |
| Chronic renal failure | 3,498 (0.3%) | 4,901 (0.5%) | 0.04 | 0.00 | 2,720 (0.2%) | 3,708 (0.5%) | 0.04 | 0.00 |
| Mild liver disease | 631 (0.0%) | 708 (0.1%) | 0.01 | 0.00 | 502 (0.0%) | 568 (0.1%) | 0.01 | 0.00 |
| Moderate-severe liver disease | 380 (0.0%) | 426 (0.0%) | 0.01 | 0.00 | 305 (0.0%) | 326 (0.0%) | 0.01 | 0.00 |
| Ulcers | 4,585 (0.4%) | 6,086 (0.6%) | 0.04 | 0.00 | 3,779 (0.3%) | 4,953 (0.6%) | 0.04 | 0.00 |
| Rheumatoid arthritis or other inflammatory polyarthropathy | 2,514 (0.2%) | 2,061 (0.2%) | 0.01 | 0.00 | 1,997 (0.2%) | 1,647 (0.2%) | 0.00 | 0.00 |
| Malignancy | 12,695 (1.0%) | 12,068 (1.3%) | 0.03 | 0.00 | 10,253 (0.9%) | 9,386 (1.1%) | 0.02 | 0.00 |
| Metastatic solid tumor | 1,094 (0.1%) | 1,108 (0.1%) | 0.01 | 0.00 | 840 (0.1%) | 802 (0.1%) | 0.01 | 0.00 |
| Medication use in the past 90 days | | | | | | | | |
| Renin–angiotensin system agent | 82,760 (6.3%) | 101,285 (10.6%) | 0.15 | 0.00 | 66,641 (6.0%) | 78,660 (9.6%) | 0.14 | 0.00 |
| Beta blocker | 46,915 (3.6%) | 57,279 (6.0%) | 0.11 | 0.00 | 37,359 (3.3%) | 43,334 (5.3%) | 0.10 | 0.00 |
| Calcium channel blocker | 124,214 (9.5%) | 158,884 (16.6%) | 0.21 | 0.00 | 101,220 (9.1%) | 125,113 (15.2%) | 0.19 | 0.00 |

*(Continued)*

**Table 1.** (Continued)

| Baseline characteristic | First dose recipients | | | | Second dose recipients | | | |
|---|---|---|---|---|---|---|---|---|
| | Before weighting | | SMD | After weighting SMD | Before weighting | | SMD | After weighting SMD |
| | Mean ± SD or N (%) | | | | Mean ± SD or N (%) | | | |
| | BNT162b2 (N = 1,308,820) | CoronaVac (N = 955,859) | | | BNT162b2 (N = 1,116,677) | CoronaVac (N = 821,560) | | |
| Diuretic | 11,455 (0.9%) | 14,847 (1.6%) | 0.06 | 0.00 | 8,929 (0.8%) | 10,863 (1.3%) | 0.05 | 0.00 |
| Nitrate | 7,081 (0.5%) | 9,705 (1.0%) | 0.05 | 0.00 | 5,550 (0.5%) | 7,200 (0.9%) | 0.05 | 0.00 |
| Lipid-lowering agent | 119,105 (9.1%) | 146,738 (15.4%) | 0.19 | 0.00 | 96,959 (8.7%) | 114,869 (14.0%) | 0.17 | 0.00 |
| Insulin | 7,284 (0.6%) | 8,406 (0.9%) | 0.04 | 0.00 | 5,527 (0.5%) | 6,031 (0.7%) | 0.03 | 0.00 |
| Antidiabetic drug | 63,170 (4.8%) | 80,903 (8.5%) | 0.15 | 0.00 | 50,403 (4.5%) | 62,001 (7.5%) | 0.13 | 0.00 |
| Antiarrhythmic drug | 579 (0.0%) | 599 (0.1%) | 0.01 | 0.00 | 445 (0.0%) | 421 (0.1%) | 0.01 | 0.00 |
| Cardiac glycoside | 739 (0.1%) | 1,049 (0.1%) | 0.02 | 0.00 | 560 (0.1%) | 689 (0.1%) | 0.01 | 0.00 |
| Oral anticoagulant | 2,924 (0.2%) | 4,028 (0.4%) | 0.03 | 0.00 | 2,167 (0.2%) | 2,772 (0.3%) | 0.03 | 0.00 |
| Parenteral anticoagulant | 396 (0.0%) | 340 (0.0%) | 0.00 | 0.00 | 261 (0.0%) | 215 (0.0%) | 0.00 | 0.00 |
| Antiplatelet | 32,943 (2.5%) | 43,862 (4.6%) | 0.11 | 0.00 | 26,140 (2.3%) | 32,890 (4.0%) | 0.09 | 0.00 |
| Antifibrinolytic or hemostatic | 4,946 (0.4%) | 3,540 (0.4%) | 0.00 | 0.00 | 4,116 (0.4%) | 2,912 (0.4%) | 0.00 | 0.00 |
| Hormonal agent (contraceptive, HRT, SERM) | 6,485 (0.5%) | 4,039 (0.4%) | 0.01 | 0.00 | 5,202 (0.5%) | 3,045 (0.4%) | 0.01 | 0.00 |
| Glucocorticoid | 6,252 (0.5%) | 4,790 (0.5%) | 0.00 | 0.00 | 4,784 (0.4%) | 3,637 (0.4%) | 0.00 | 0.00 |
| Antidepressant | 31,259 (2.4%) | 26,354 (2.8%) | 0.02 | 0.00 | 25,771 (2.3%) | 21,280 (2.6%) | 0.02 | 0.00 |
| Bevacizumab | 32 (0.0%) | 23 (0.0%) | 0.00 | 0.00 | 24 (0.0%) | 13 (0.0%) | 0.00 | 0.00 |
| Tranexamic acid | 4,921 (0.4%) | 3,526 (0.4%) | 0.00 | 0.00 | 4,093 (0.4%) | 2,905 (0.4%) | 0.00 | 0.00 |
| Intravenous immunoglobulin | 47 (0.0%) | 28 (0.0%) | 0.00 | 0.00 | 33 (0.0%) | 15 (0.0%) | 0.00 | 0.00 |
| NSAID | 72,371 (5.5%) | 54,715 (5.7%) | 0.01 | 0.00 | 62,948 (5.6%) | 46,969 (5.7%) | 0.00 | 0.00 |
| Drug for gout | 11,084 (0.8%) | 12,782 (1.3%) | 0.05 | 0.00 | 9,167 (0.8%) | 10,295 (1.3%) | 0.04 | 0.00 |
| Antiepileptic drug | 12,728 (1.0%) | 10,132 (1.1%) | 0.01 | 0.00 | 10,171 (0.9%) | 7,498 (0.9%) | 0.00 | 0.00 |
| Drug for control of epilepsy | 12,722 (1.0%) | 10,125 (1.1%) | 0.01 | 0.00 | 10,163 (0.9%) | 7,493 (0.9%) | 0.00 | 0.00 |
| Antiviral drug | 8,342 (0.6%) | 8,192 (0.9%) | 0.03 | 0.00 | 6,845 (0.6%) | 6,587 (0.8%) | 0.02 | 0.00 |
| Antibacterial drug | 38,297 (2.9%) | 28,903 (3.0%) | 0.01 | 0.00 | 31,342 (2.8%) | 23,029 (2.8%) | 0.00 | 0.00 |
| Immunosuppressant | 3,634 (0.3%) | 2,336 (0.2%) | 0.01 | 0.00 | 2,701 (0.2%) | 1,772 (0.2%) | 0.01 | 0.00 |
| Adrenaline | 91 (0.0%) | 132 (0.0%) | 0.01 | 0.00 | 60 (0.0%) | 77 (0.0%) | 0.00 | 0.00 |
| Oral contraceptive | 1,162 (0.1%) | 388 (0.0%) | 0.02 | 0.00 | 943 (0.1%) | 320 (0.0%) | 0.02 | 0.00 |
| HRT | 2,947 (0.2%) | 1,711 (0.2%) | 0.01 | 0.00 | 2,407 (0.2%) | 1,374 (0.2%) | 0.01 | 0.00 |
| Hormonal agent in malignant disease | 2,463 (0.2%) | 1,960 (0.2%) | 0.00 | 0.00 | 1,912 (0.2%) | 1,372 (0.2%) | 0.00 | 0.00 |

AESI, adverse event of special interest; HRT, hormone replacement therapy; NSAID, non-steroidal anti-inflammatory drug; SD, standard deviation; SERM, selective estrogen receptor modulator; SMD, standardized mean difference.

[a]Previous SARS-CoV-2 infection was defined as ever having a positive result on the SARS-CoV-2 reverse transcription polymerase chain reaction (RT-PCR) test before COVID-19 vaccination.

100,000 person-years), anaphylaxis (67 and 212 per 100,000 person-years), MI (148 and 107 per 100,000 person-years), and sleeping disturbance or disorder (146 and 142 per 100,000 person-years), consistent with results of the first dose. In contrast to the significant differences observed for Bell palsy (higher incidence rate for CoronaVac) and anaphylaxis and sleeping disturbance or disorder (higher incidence rates for BNT162b2) following the first dose, none of the AESIs investigated had demonstrated significant differences in incidence rate for second dose CoronaVac versus BNT162b2 (Table 3).

**Table 2. Cumulative incidence and crude incidence rate of AESIs among first dose recipients of CoronaVac or BNT162b2 from 23 February to 9 September 2021 in Hong Kong SAR, China.**

| Outcome | BNT162b2 (N = 1,308,820) | | | | CoronaVac (N = 955,859) | | | |
|---|---|---|---|---|---|---|---|---|
| | Cumulative incidence (events/100,000 doses) | | Crude incidence rate (events/100,000 person-years) | | Cumulative incidence (events/100,000 doses) | | Crude incidence rate (events/100,000 person-years) | |
| | Cases with event | Rate | Estimate | 95% CI | Cases with event | Rate | Estimate | 95% CI |
| Overall AESIs | 850 | 64.94 | 1,162 | 1,085.02, 1,242.63 | 861 | 90.08 | 1,566 | 1,463.42, 1,674.52 |
| Guillain-Barré syndrome | 2 | 0.15 | 2.73 | 0.33, 9.87 | — | — | — | — |
| Acute disseminated encephalomyelitis | — | — | — | — | — | — | — | — |
| Sleeping disturbance or disorder | 122 | 9.32 | 166.71 | 138.44, 199.05 | 72 | 7.53 | 130.93 | 102.45, 164.89 |
| Acute aseptic arthritis | 28 | 2.14 | 38.26 | 25.42, 55.30 | 34 | 3.56 | 61.83 | 42.82, 86.40 |
| Type 1 diabetes | 1 | 0.08 | 1.37 | 0.03, 7.61 | 1 | 0.10 | 1.82 | 0.05, 10.13 |
| (Idiopathic) thrombocytopenia | 10 | 0.76 | 13.66 | 6.55, 25.13 | 5 | 0.52 | 9.09 | 2.95, 21.22 |
| Subacute thyroiditis | 1 | 0.08 | 1.37 | 0.03, 7.61 | — | — | — | — |
| Microangiopathy | 3 | 0.23 | 4.10 | 0.85, 11.98 | 1 | 0.10 | 1.82 | 0.05, 10.13 |
| Heart failure | 21 | 1.60 | 28.69 | 17.76, 43.86 | 54 | 5.65 | 98.20 | 73.77, 128.12 |
| Stress cardiomyopathy | — | — | — | — | — | — | — | — |
| Arrhythmia | 123 | 9.40 | 168.08 | 139.69, 200.54 | 130 | 13.60 | 236.41 | 197.52, 280.71 |
| Carditis | 10 | 0.76 | 13.66 | 6.55, 25.13 | 3 | 0.31 | 5.46 | 1.13, 15.94 |
| Thromboembolism | 212 | 16.20 | 289.70 | 252.01, 331.43 | 237 | 24.79 | 431.01 | 377.88, 489.52 |
| Coronary artery disease | 115 | 8.79 | 157.14 | 129.74, 188.63 | 149 | 15.59 | 270.96 | 229.20, 318.13 |
| Myocardial infarction | 76 | 5.81 | 103.85 | 81.82, 129.98 | 99 | 10.36 | 180.03 | 146.32, 219.18 |
| Venous thromboembolism | 16 | 1.22 | 21.86 | 12.50, 35.50 | 10 | 1.05 | 18.18 | 8.72, 33.44 |
| Arterial thromboembolism | 104 | 7.95 | 142.11 | 116.12, 172.19 | 134 | 14.02 | 243.68 | 204.17, 288.61 |
| Hemorrhagic disease | 43 | 3.29 | 58.76 | 42.52, 79.14 | 47 | 4.92 | 85.47 | 62.80, 113.65 |
| Single organ cutaneous vasculitis | 2 | 0.15 | 2.73 | 0.33, 9.87 | — | — | — | — |
| Acute liver injury | 13 | 0.99 | 17.76 | 9.46, 30.38 | 11 | 1.15 | 20.00 | 9.99, 35.79 |
| Acute kidney injury | 95 | 7.26 | 129.81 | 105.03, 158.69 | 120 | 12.55 | 218.22 | 180.92, 260.93 |
| Acute pancreatitis | 17 | 1.30 | 23.23 | 13.53, 37.19 | 15 | 1.57 | 27.28 | 15.27, 44.99 |
| Generalized convulsion | 57 | 4.36 | 77.89 | 58.99, 100.91 | 42 | 4.39 | 76.37 | 55.04, 103.24 |
| Meningoencephalitis | 9 | 0.69 | 12.30 | 5.62, 23.34 | 1 | 0.10 | 1.82 | 0.05, 10.13 |
| Transverse myelitis | — | — | — | — | — | — | — | — |
| Bell palsy | 22 | 1.68 | 30.06 | 18.84, 45.51 | 37 | 3.87 | 67.28 | 47.37, 92.74 |
| Acute respiratory distress syndrome | 16 | 1.22 | 21.86 | 12.50, 35.50 | 16 | 1.67 | 29.09 | 16.63, 47.25 |
| Erythema multiforme | 2 | 0.15 | 2.73 | 0.33, 9.87 | 1 | 0.10 | 1.82 | 0.05, 10.13 |
| Chilblain-like lesions | — | — | — | — | — | — | — | — |
| Anosmia, ageusia | — | — | — | — | — | — | — | — |
| Anaphylaxis | 27 | 2.06 | 376 | 248.11, 547.77 | 7 | 0.73 | 134 | 53.73, 275.35 |
| Multisystem inflammatory syndrome | — | — | — | — | — | — | — | — |
| Sudden death | 9 | 0.69 | 12.30 | 5.62, 23.34 | 13 | 1.36 | 23.64 | 12.59, 40.42 |
| Rhabdomyolysis | 7 | 0.53 | 9.56 | 3.85, 19.71 | 6 | 0.63 | 10.91 | 4.00, 23.75 |
| All-cause mortality | 42 | 3.21 | 57.39 | 41.36, 77.57 | 53 | 5.54 | 96.38 | 72.19, 126.06 |

AESI, adverse event of special interest; CI, confidence interval. The cumulative incidence and crude incidence rate of some AESIs were not reported because the number of events for that AESI was 0.

## Sensitivity analysis

Results of the sensitivity analysis excluding vaccinated individuals with previous SARS-CoV-2 infection are presented in S3 Table. Overall, they were consistent with those of the main

**Table 3. Incidence rate ratio of AESIs among first dose and second dose recipients of CoronaVac versus BNT162b2 as the reference category (after weighting).**

| Outcome | First dose recipients | | | Second dose recipients | | |
|---|---|---|---|---|---|---|
| | IRR | 95% CI | *p*-Value | IRR | 95% CI | *p*-Value |
| Overall AESIs | 0.98 | 0.89, 1.08 | 0.703 | 0.97 | 0.87, 1.08 | 0.545 |
| Sleeping disturbance or disorder | 0.66 | 0.49, 0.89 | 0.006 | 0.91 | 0.66, 1.27 | 0.586 |
| Acute aseptic arthritis | 1.33 | 0.79, 2.24 | 0.281 | 1.13 | 0.68, 1.90 | 0.636 |
| (Idiopathic) thrombocytopenia | 0.72 | 0.24, 2.16 | 0.555 | — | — | — |
| Heart failure | 1.59 | 0.94, 2.70 | 0.083 | 1.45 | 0.76, 2.74 | 0.256 |
| Arrhythmia | 0.90 | 0.69, 1.16 | 0.412 | 0.96 | 0.72, 1.27 | 0.764 |
| Thromboembolism | 0.94 | 0.78, 1.14 | 0.543 | 0.93 | 0.75, 1.15 | 0.501 |
| Coronary artery disease | 1.13 | 0.88, 1.44 | 0.354 | 1.13 | 0.86, 1.47 | 0.376 |
| Myocardial infarction | 1.18 | 0.87, 1.60 | 0.287 | 0.91 | 0.65, 1.28 | 0.603 |
| Venous thromboembolism | 0.58 | 0.26, 1.29 | 0.180 | — | — | — |
| Arterial thromboembolism | 1.02 | 0.79, 1.34 | 0.857 | 1.12 | 0.84, 1.50 | 0.446 |
| Hemorrhagic disease | 1.02 | 0.66, 1.57 | 0.932 | 0.63 | 0.38, 1.04 | 0.070 |
| Acute liver injury | 1.31 | 0.57, 2.98 | 0.523 | 0.59 | 0.21, 1.66 | 0.317 |
| Acute kidney injury | 1.09 | 0.82, 1.45 | 0.545 | 0.82 | 0.60, 1.11 | 0.194 |
| Acute pancreatitis | 1.08 | 0.51, 2.30 | 0.837 | 1.62 | 0.81, 3.24 | 0.173 |
| Generalized convulsion | 1.41 | 0.92, 2.17 | 0.117 | 0.73 | 0.43, 1.25 | 0.250 |
| Bell palsy | 1.95 | 1.12, 3.41 | 0.018 | 0.88 | 0.51, 1.51 | 0.641 |
| Acute respiratory distress syndrome | 0.81 | 0.40, 1.66 | 0.571 | 1.27 | 0.76, 2.13 | 0.360 |
| Anaphylaxis | 0.34 | 0.14, 0.79 | 0.012 | — | — | — |
| Sudden death | 0.97 | 0.39, 2.38 | 0.944 | 1.80 | 0.69, 4.73 | 0.232 |
| Rhabdomyolysis | 1.48 | 0.46, 4.79 | 0.513 | — | — | — |
| All-cause mortality | 0.96 | 0.63, 1.48 | 0.868 | 0.85 | 0.51, 1.40 | 0.516 |

AESI, adverse event of special interest; CI, confidence interval; IRR, incidence rate ratio. IRR was estimated only when there were at least 5 events for a specific AESI in both the BNT162b2 and CoronaVac groups.

analysis, suggesting a comparable safety profile between the 2 vaccines, alongside a significantly higher incidence rate of Bell palsy, but lower incidence rates of anaphylaxis and sleeping disturbance or disorder, following first dose CoronaVac versus BNT162b2 vaccination.

## Discussion

In this territory-wide cohort study, the risks of AESIs and all-cause mortality in temporal association with the first and second doses of CoronaVac and BNT162b2 were observed to be very low. Despite the slightly higher crude incidence rates of overall AESIs among CoronaVac recipients than among BNT162b2 recipients, the incidence rates of overall AESIs and all-cause mortality were comparable between the 2 vaccines after weighting, except for a significantly higher incidence rate of Bell palsy identified over the 21 days of post-vaccination follow-up for first dose CoronaVac versus BNT162b2 recipients, and lower incidence rates of anaphylaxis and sleeping disturbance or disorder observed for first dose CoronaVac versus BNT162b2 recipients. Notably, these significant associations did not persist during the follow-up period of the second dose vaccination. Also, the wider confidence intervals in the results of the sensitivity analysis for acute kidney injury suggested the difference was not significant, possibly caused by the small number of events following COVID-19 vaccination.

As with any vaccines and pharmaceutical products, anaphylaxis is a potential adverse event of major concern given its severe and life-threatening nature. Regarding the 2 different vaccine

**Table 4. Cumulative incidence and crude incidence rate of AESIs among second dose recipients of CoronaVac or BNT162b2 from 23 February to 9 September 2021 in Hong Kong SAR, China.**

| Outcome | BNT162b2 (N = 1,116,677) | | | | CoronaVac (N = 821,560) | | | |
|---|---|---|---|---|---|---|---|---|
| | Cumulative incidence (events/100,000 doses) | | Crude incidence rate (events/100,000 person-years) | | Cumulative incidence (events/100,000 doses) | | Crude incidence rate (events/100,000 person-years) | |
| | Cases with event | Rate | Estimate | 95% CI | Cases with event | Rate | Estimate | 95% CI |
| Overall AESIs | 733 | 65.64 | 1,141 | 1,060.15, 1,226.98 | 720 | 87.64 | 1,524 | 1,414.58, 1,639.37 |
| Guillain-Barré syndrome | — | — | — | — | — | — | — | — |
| Acute disseminated encephalomyelitis | — | — | — | — | 1 | 0.12 | 2.12 | 0.05, 11.79 |
| Sleeping disturbance or disorder | 91 | 8.15 | 141.65 | 114.05, 173.91 | 69 | 8.40 | 145.98 | 113.59, 184.75 |
| Acute aseptic arthritis | 31 | 2.78 | 48.25 | 32.79, 68.49 | 31 | 3.77 | 65.59 | 44.56, 93.09 |
| Type 1 diabetes | 2 | 0.18 | 3.11 | 0.38, 11.25 | — | — | — | — |
| (Idiopathic) thrombocytopenia | 7 | 0.63 | 10.90 | 4.38, 22.45 | 1 | 0.12 | 2.12 | 0.05, 11.79 |
| Subacute thyroiditis | — | — | — | — | — | — | — | — |
| Microangiopathy | 2 | 0.18 | 3.11 | 0.38, 11.25 | 1 | 0.12 | 2.12 | 0.05, 11.79 |
| Heart failure | 15 | 1.34 | 23.35 | 13.07, 38.51 | 30 | 3.65 | 63.47 | 42.82, 90.61 |
| Stress cardiomyopathy | — | — | — | — | — | — | — | — |
| Arrhythmia | 101 | 9.04 | 157.21 | 128.05, 191.03 | 104 | 12.66 | 220.04 | 179.79, 266.62 |
| Carditis | 16 | 1.43 | 24.90 | 14.23, 40.44 | 2 | 0.24 | 4.23 | 0.51, 15.28 |
| Thromboembolism | 171 | 15.31 | 266.18 | 227.78, 309.21 | 182 | 22.15 | 385.09 | 331.17, 445.28 |
| Coronary artery disease | 101 | 9.04 | 157.21 | 128.05, 191.03 | 128 | 15.58 | 270.82 | 225.94, 322.00 |
| Myocardial infarction | 69 | 6.18 | 107.40 | 83.57, 135.93 | 70 | 8.52 | 148.10 | 115.45, 187.12 |
| Venous thromboembolism | 10 | 0.90 | 15.57 | 7.46, 28.62 | 4 | 0.49 | 8.46 | 2.31, 21.67 |
| Arterial thromboembolism | 82 | 7.34 | 127.64 | 101.51, 158.43 | 111 | 13.51 | 234.85 | 193.20, 282.82 |
| Hemorrhagic disease | 37 | 3.31 | 57.59 | 40.55, 79.38 | 29 | 3.53 | 61.35 | 41.09, 88.12 |
| Single organ cutaneous vasculitis | 1 | 0.09 | 1.56 | 0.04, 8.67 | — | — | — | — |
| Acute liver injury | 11 | 0.99 | 17.12 | 8.55, 30.64 | 6 | 0.73 | 12.69 | 4.66, 27.63 |
| Acute kidney injury | 91 | 8.15 | 141.65 | 114.05, 173.91 | 83 | 10.10 | 175.60 | 139.87, 217.69 |
| Acute pancreatitis | 14 | 1.25 | 21.79 | 11.91, 36.56 | 21 | 2.56 | 44.43 | 27.50, 67.92 |
| Generalized convulsion | 44 | 3.94 | 68.49 | 49.76, 91.94 | 22 | 2.68 | 46.54 | 29.17, 70.47 |
| Meningoencephalitis | 2 | 0.18 | 3.11 | 0.38, 11.25 | 1 | 0.12 | 2.12 | 0.05, 11.79 |
| Transverse myelitis | — | — | — | — | 1 | 0.12 | 2.12 | 0.05, 11.79 |
| Bell palsy | 34 | 3.04 | 52.92 | 36.65, 73.95 | 24 | 2.92 | 50.78 | 32.53, 75.55 |
| Acute respiratory distress syndrome | 25 | 2.24 | 38.91 | 25.18, 57.44 | 38 | 4.63 | 80.40 | 56.89, 110.35 |
| Erythema multiforme | 2 | 0.18 | 3.11 | 0.38, 11.25 | 2 | 0.24 | 4.23 | 0.51, 15.28 |
| Chilblain-like lesions | — | — | — | — | — | — | — | — |
| Anosmia, ageusia | 1 | 0.09 | 1.56 | 0.04, 8.67 | — | — | — | — |
| Anaphylaxis | 13 | 1.16 | 212 | 113.12, 363.30 | 3 | 0.37 | 67 | 13.74, 194.74 |
| Multisystem inflammatory syndrome | 1 | 0.09 | 1.56 | 0.04, 8.67 | — | — | — | — |
| Sudden death | 8 | 0.72 | 12.45 | 5.38, 24.54 | 13 | 1.58 | 27.50 | 14.64, 47.03 |
| Rhabdomyolysis | 4 | 0.36 | 6.23 | 1.70, 15.94 | 6 | 0.73 | 12.69 | 4.66, 27.63 |
| All-cause mortality | 34 | 3.04 | 52.92 | 36.65, 73.95 | 33 | 4.02 | 69.82 | 48.06, 98.05 |

AESI, adverse event of special interest; CI, confidence interval. The cumulative incidence and crude incidence rate of some AESIs were not reported because the number of events for that AESI was 0.

platforms in this study, the inactivated virus and fragments, as well as the adjuvant aluminum hydroxide, of CoronaVac have been suggested as possible allergens, while a possible allergen of BNT162b2 could be its polyethylene glycol (PEG) excipient [56,57]. Proposed mechanisms of

allergic reactions to these 2 COVID-19 vaccines include IgE- and non-IgE-mediated reactions, mast cell and complement activation, and delayed hypersensitivity [56,57]. Compared to an estimated rate of anaphylaxis of 2.2 per million doses of CoronaVac [57], the administration of BNT162b2 was associated with higher rates of anaphylaxis, ranging from 4.7 to 13.7 per million doses, which could vary depending on the priority groups for vaccination [15,58,59]. Consistent with previous observations that anaphylaxis was clinically manifested more frequently after the first dose of either vaccine [57,59], it remains to be determined if these events would be considered anaphylaxis with previous sensitization to specific components of the vaccines, or anaphylactoid reactions occurring on their first exposure [60]. In line with the current literature, our results suggested a higher incidence rate of anaphylaxis after first dose BNT162b2 versus CoronaVac vaccination. While no fatal anaphylaxis with either vaccine has been reported so far, several cases of mortality in close temporal relationship with COVID-19 vaccination have raised major public concern, such as death possibly due to intracranial hemorrhage or myocarditis following BNT162b2 vaccination [61,62]. Nevertheless, autopsy and postmortem findings have not established a causal link between COVID-19 vaccination and death, given the multimorbidity status and relatively older ages of those who died, where decompensation could result from common side effects of immunization [63].

With respect to several neurological AESIs, our study was able to replicate the results of a case series and nested case–control study demonstrating a significantly higher incidence rate and odds of Bell palsy following the administration of CoronaVac versus BNT162b2, especially after the first vaccine dose [28]. Despite an imbalance in the number of cases found in COVID-19 mRNA vaccine trials [64], the risk of Bell palsy was not significantly increased with BNT162b2 vaccination compared to unvaccinated controls or across different risk intervals in surveillance studies [14,15,21,22]. Apart from an association with herpes zoster infection, autoimmunity might play a role in the development of Bell palsy, hence calling for an investigation into the immunomodulatory effects of viral antigens and adjuvants of different vaccine platforms [14,22,28].

Over the last decade, autoimmune/inflammatory syndrome induced by adjuvants (ASIA) has been proposed to collectively include a variety of post-vaccination phenomena associated with adjuvants and subsequent clinical manifestations of autoimmune diseases, for example, GBS, TM, and subacute thyroiditis [65,66]. In this study, IRR estimation for GBS, TM, and subacute thyroiditis were not performed owing to the very few cases recorded in our dataset. In fact, while both GBS and TM have not been investigated among CoronaVac recipients, BNT162b2 vaccination has not been shown to be associated with an elevated risk of either AESI, as other risk factors such as concomitant infections and autoimmunity might play a more prominent role in these 2 neurological disorders [15,21,67]. Regarding subacute thyroiditis, case reports have suggested that the aluminum adjuvant of CoronaVac, and cross-recognition between the SARS-CoV-2 spike protein of BNT162b2 and thyroid cell antigens, may induce this endocrine disorder [9,68]. Regarding the higher incidence rate of sleeping disturbance or disorder following a first BNT162b2 dose than following a first CoronaVac dose, there is a lack of evidence suggesting a correlation with initial reactogenicity. Based on our best knowledge and the current literature, we are unable to identify any biological or immunological mechanisms to explain sleeping disturbances following COVID-19 vaccination; hence, further investigation is needed.

Since the rollout of mass vaccination and continuous safety monitoring, numerous studies have concluded that BNT162b2 is potentially associated with an increased risk of myocarditis, especially among males under the age of 30 years and within the first week following second dose vaccination [14–16,69]. Despite the small number of cases and without an estimation of IRR in this study, higher crude incidence rates of carditis were consistently demonstrated

among BNT162b2 versus CoronaVac recipients for both vaccine doses, and in particular after the second dose. The small number of carditis cases in this study may be attributed to our exclusion of vaccinated individuals under the age of 18 years old, as a significantly increased risk of acute myocarditis or pericarditis has been identified among male adolescents following BNT162b2 vaccination in local studies, especially after the second dose [18,70,71]. Based on the above observations, several mechanisms have been proposed regarding the development of myocarditis and pericarditis following COVID-19 mRNA vaccination, for instance, robust antibody response in younger recipients, hypersensitivity upon second exposure to the viral antigen, as well as cardiac inflammation mediated through cross-reactivity, molecular mimicry, or bystander activation [16,72]. Regarding arrhythmia, CAD, and MI, no significant differences in their incidence rates were identified between CoronaVac and BNT162b2 recipients in our cohort. For the mRNA vaccine, previous studies have shown comparable risks of these cardiac events in vaccinated individuals and unvaccinated controls, and across different risk intervals [14,15,23]; their incidences following the inactivated vaccine are yet to be evaluated, apart from 1 case of CAD and 1 case of coronary atherosclerosis reported in 1 of the clinical trials for CoronaVac [4].

Among the numerous COVID-19 vaccines administered worldwide, adenoviral vector vaccines have been associated with rare AESI of thromboembolism, thrombocytopenia, and hemorrhagic events [73,74], while the relationship of these AESIs with mRNA vaccines remain inconclusive. Studies have found no association between BNT162b2 vaccination and thromboembolic events compared to unvaccinated controls, a marginally increased risk of venous thromboembolism with COVID-19 mRNA vaccines during days 1–21 versus days 22–42 after vaccination, as well as increased IRR for arterial thromboembolism, cerebral venous sinus thrombosis, and ischemic and hemorrhagic stroke during post-vaccination follow-up in comparison to baseline periods of self-controlled case series [14,15,21,23]. While these events could be mediated by systemic inflammation in response to vaccination and immune responses of free-floating spike proteins interacting with angiotensin-converting enzyme 2 (ACE2) to promote platelet aggregation, BNT162b2 vaccination does not seem to favor these outcomes, nor does the inactivated vaccine CoronaVac [73–75]. Overall, the IRRs for the above AESIs were not statistically significant comparing between CoronaVac and BNT162b2 recipients of either dose.

The strengths of this population-based study included the large sample size of vaccinated individuals, and capturing the occurrence of rare AESIs after both the first and second doses of COVID-19 vaccines. Also, the small proportion of vaccinated individuals with previous SARS-CoV-2 infection would have minimized any effects of long COVID on specific AESIs. This territory-wide cohort study provided substantial evidence comparing the incidence rates of various AESIs and all-cause mortality following vaccination with an inactivated COVID-19 vaccine (CoronaVac) and an mRNA vaccine (BNT162b2, Comirnaty), which could be useful in safety monitoring comparing different vaccine platforms, and complementary to the current literature comparing vaccinated individuals to unvaccinated controls. Nevertheless, several study limitations of this analysis should be addressed. First, similar to all epidemiological studies, even though the demographic and clinical characteristics of vaccine recipients were balanced with PS weighting at baseline, residual confounding could still exist. Second, certain events might have been missed if patients were managed under the private healthcare system and without data linkage to the Hospital Authority; however, there is no reason why recipients of one type of vaccine would be more likely to use private care than recipients of the other type, and, consequently, use of private care is unlikely to affect the interpretation of our results. Third, the observed incidences of some AESIs were very low in this cohort, and our study might be underpowered to draw any definite conclusions about these AESIs; however, it is also

reassuring that recipients of either vaccine were very unlikely to experience these serious AESIs. Lastly, our electronic medical records did not allow access to the free text, or contact with doctors or patients; therefore, we were unable to assess cases using the Brighton case definition of anaphylaxis and level of certainty [76].

By evaluating and comparing the safety profile of CoronaVac against BNT162b2 (Comirnaty), the first FDA-approved SARS-CoV-2 vaccine for COVID-19 prevention, this study could help inform the choice of inactivated COVID-19 vaccines, mainly administered in low- and middle-income countries with large populations, in comparison to the safety of mRNA vaccines. Despite the slightly higher crude incidence rates of overall AESIs among CoronaVac recipients than among BNT162b2 recipients, the 2 vaccine platforms had a similar safety profile after adjusting for baseline characteristics. As currently done for mRNA vaccines, clinical data and observational reports of COVID-19 vaccines should continue to be made publicly available for all vaccine platforms in a timely manner [77,78], as a means of facilitating the continuous monitoring of vaccine safety via pharmacovigilance and pharmacoepidemiology. Further research comparing the safety and efficacy of different novel COVID-19 vaccine platforms—as well as homologous versus heterologous boosting regimens and different numbers of booster doses administered—is urgently needed.

In this territory-wide cohort study, the incidence rates of AESIs (cumulative incidence rate of 0.06%–0.09%) and all-cause mortality following the first and second doses of CoronaVac and BNT162b2 vaccination were very low. The safety profiles of CoronaVac (inactivated COVID-19 vaccine) and BNT162b2 (mRNA vaccine) were generally comparable regarding various AESIs and all-cause mortality following the first and second doses, except for a significantly higher incidence rate of Bell palsy, and lower incidence rates of anaphylaxis and sleeping disturbance or disorder, among first dose CoronaVac recipients than among first dose BNT162b2 recipients. Long-term surveillance of the safety profile of COVID-19 vaccines should continue.

## Supporting information

**S1 Checklist. Strengthening the Reporting of Observational Studies in Epidemiology (STROBE) checklist.**
(PDF)

**S1 Table. Definitions of adverse events of special interest (AESIs).**
(PDF)

**S2 Table. Goodness-of-fit test for Poisson regression models.**
(PDF)

**S3 Table. Sensitivity analysis of the study outcomes excluding people with previous SARS-CoV-2 infection.**
(PDF)

**S1 Fig. The observed propensity score weighting distribution after truncation of the 1st and 99th percentiles, by the first and second vaccine dose.** IPTW, inverse probability of treatment weighting.
(PDF)

**S2 Fig. Distribution of propensity score density by the first and second vaccine dose before and after weighting.**
(PDF)

**S3 Fig. Distribution of the occurrence of sleeping disturbance or disorder within 21 days after the first dose of BNT162b2.**
(PDF)

**S1 Protocol. Study protocol.**
(PDF)

## Acknowledgments

The authors thank the Hospital Authority and the Department of Health for the generous provision of data for this study.

## Author Contributions

**Conceptualization:** Carlos King Ho Wong, Ian Chi Kei Wong.

**Data curation:** Carlos King Ho Wong.

**Formal analysis:** Carlos King Ho Wong, Xi Xiong, Ivan Chi Ho Au.

**Funding acquisition:** Carlos King Ho Wong, Ian Chi Kei Wong.

**Investigation:** Carlos King Ho Wong, Kristy Tsz Kwan Lau, Xi Xiong, Ivan Chi Ho Au, Ian Chi Kei Wong.

**Methodology:** Carlos King Ho Wong, Francisco Tsz Tsun Lai, Eric Yuk Fai Wan, Xue Li, Esther Wai Yin Chan.

**Project administration:** Carlos King Ho Wong, Ian Chi Kei Wong.

**Supervision:** Carlos King Ho Wong, Ian Chi Kei Wong.

**Validation:** Carlos King Ho Wong, Francisco Tsz Tsun Lai, Eric Yuk Fai Wan, Celine Sze Ling Chui, Xue Li, Esther Wai Yin Chan, Le Gao, Franco Wing Tak Cheng, Sydney Chi Wai Tang, Ian Chi Kei Wong.

**Visualization:** Xi Xiong, Ivan Chi Ho Au.

**Writing – original draft:** Carlos King Ho Wong, Kristy Tsz Kwan Lau, Xi Xiong, Ivan Chi Ho Au.

**Writing – review & editing:** Francisco Tsz Tsun Lai, Eric Yuk Fai Wan, Celine Sze Ling Chui, Xue Li, Esther Wai Yin Chan, Le Gao, Franco Wing Tak Cheng, Sydney Chi Wai Tang, Ian Chi Kei Wong.

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
