## [Editor Report · Decision Letter 0]

7 Jan 2022

Dear Dr Wong, 

Thank you for submitting your manuscript entitled "Adverse Events of Special Interest and mortality following the mRNA (BNT162b2) and inactivated (CoronaVac) SARS-CoV-2 vaccines" for consideration by PLOS Medicine.

Your manuscript has now been evaluated by the PLOS Medicine editorial staff and I am writing to let you know that we would like to send your submission out for external peer review.

Please re-submit your manuscript within two working days, i.e. by Jan 11 2022 11:59PM.

Kind regards,

Caitlin Moyer, Ph.D.

Associate Editor

PLOS Medicine

---

## [Decision Letter · Decision Letter 1]

2 Mar 2022

Dear Dr. Wong,

Thank you very much for submitting your manuscript "Adverse Events of Special Interest and mortality following the mRNA (BNT162b2) and inactivated (CoronaVac) SARS-CoV-2 vaccines" (PMEDICINE-D-21-05271R1) for consideration at PLOS Medicine. 

Your paper was evaluated by a senior editor and discussed among all the editors here. It was also discussed with an academic editor with relevant expertise, and sent to three independent reviewers, including a statistical reviewer. The reviews are appended at the bottom of this email and any accompanying reviewer attachments can be seen via the link below:

[LINK]

In light of these reviews, I am afraid that we will not be able to accept the manuscript for publication in the journal in its current form, but we would like to consider a revised version that addresses the reviewers' and editors' comments. Obviously we cannot make any decision about publication until we have seen the revised manuscript and your response, and we plan to seek re-review by one or more of the reviewers. 

We expect to receive your revised manuscript by Mar 23 2022 11:59PM. Please email us (plosmedicine@plos.org) if you have any questions or concerns.

We look forward to receiving your revised manuscript. 

Sincerely,

Caitlin Moyer, Ph.D.

Associate Editor

PLOS Medicine

plosmedicine.org

1. From the academic editor: Please include additional high-level contextualization of the data in the Abstract, Results, and Discussion sections. Specifically, please discuss that less than 1% of vaccine doses were associated with an AESI, with the overall AESI rate slightly higher in CoronaVac compared to BNT162b2.

2. Title: Please revise your title according to PLOS Medicine's style. Your title must be nondeclarative and not a question. It should begin with main concept if possible. "Effect of" should be used only if causality can be inferred, i.e., for an RCT. Please place the study design ("A randomized controlled trial," "A retrospective study," "A modelling study," etc.) in the subtitle (ie, after a colon).

3. Data availability statement: Please clarify the availability of vaccination record data owned by the Department of Health in Hong Kong. If the data are owned by a third party but freely available upon request, please note this and state the owner of the data set and contact information for data requests (web or email address). Note that a study author cannot be the contact person for the data.If the data are not freely available, please describe briefly the ethical, legal, or contractual restriction that prevents you from sharing it. Please also include an appropriate contact (web or email address) for inquiries (again, this cannot be a study author).

4. Competing Interests: Several authors report research funding or grants from Pfizer, the makers of the BNT162b2 vaccine. For authors with ties to industry, please indicate whether any of the interests has a financial stake in the results of the current study.

5. Abstract: Background: Please provide an additional sentence of context of why the study is important.

6. Abstract: Methods and Findings: Please also present some of the specific AESI and mortality results for first and second doses of each vaccine (e.g rates overall for AESI and mortality, and for some of the more common AESIs such as thromboembolism). Please also present results for some of the AESIs for which the two vaccines were generally comparable and more frequently reported.

7. Abstract: Conclusion: We suggest beginning the first sentence with: "In this study, we observed ..." and we suggest including a sentence to address the study implications.

8. Author summary: At this stage, we ask that you include a short, non-technical Author Summary of your research to make findings accessible to a wide audience that includes both scientists and non-scientists. The Author Summary should immediately follow the Abstract in your revised manuscript. This text is subject to editorial change and should be distinct from the scientific abstract. Please see our author guidelines for more information: https://journals.plos.org/plosmedicine/s/revising-your-manuscript#loc-author-summary

9. Throughout: In-text citations: Please place in-text citations within square brackets, placed before the sentence punctuation, for example [1]. Where multiple references are cited, please do not include spaces within brackets.

10. Methods: Please ensure that the study is reported according to the STROBE guideline. Please add the following statement, or similar, to the Methods: "This study is reported as per the Strengthening the Reporting of Observational Studies in Epidemiology (STROBE) guideline (S1 Checklist)."

11. Methods: Did your study have a prospective protocol or analysis plan? Please state this (either way) early in the Methods section.

12. Methods: Line 131: Please clarify as (Figure 1) here.

13. Methods: Line 172-175: Please explain here exactly how the AESI outcomes were adapted from the WHO Global Advisory Committee on Vaccine Safety list of AESI (e.g. the specific rationale for adding to the conditions listed in Table 4 of reference 28).

14. Methods: Line 185: Previous SARS-CoV-2 infection: Please provide some information as to how this covariate was assessed. In Table 1, this variable is indicated by “COVID-19 survivor” and it may help to provide a clear description of the definition of this measure.

15. Methods: Line 197: Please provide more specific information on how covariates “were taken into account” here.

16. Methods: Line 203: Please clarify how “history of AESI” is defined (for example, does this indicate a history of any of the conditions listed in Table S1, over the same time period over which other covariates/comorbidities were considered?).

17. Results: Line 272: “Nevertheless, none of the AESI investigated had demonstrated significant differences in the incidence rate comparing between second dose CoronaVac and BNT162b2 (Table 4).” It may be useful here to point out the results for comparisons for Bell’s palsy, anaphylaxis, and sleeping disturbance/disorder (those AESIs that differed significantly for the first dose).

18. Discussion: Please present and organize the Discussion as follows: a short, clear summary of the article's findings; what the study adds to existing research and where and why the results may differ from previous research; strengths and limitations of the study; implications and next steps for research, clinical practice, and/or public policy; one-paragraph conclusion.

19. References: Please use the "Vancouver" style for reference formatting, and please check each reference for NLM abbreviations for journal titles. Please see our website for other reference guidelines https://journals.plos.org/plosmedicine/s/submission-guidelines#loc-references

20. Table 1: Please provide numbers in addition to percentages. Please clarify if “Overall AESI” under “Pre-existing comorbidities” refers to the history of AESI conditions, prior to vaccination.

21. Tables 2 and 3: Please describe in the legend, the rationale for unreported incidence (for example, for ADEM).

22. STROBE Checklist: Thank you for including the STROBE Checklist. Please revise the checklist, referring to locations within the text by section and paragraph numbers, rather than page numbers (for example, Methods, Paragraph 1).

Comments from the reviewers:

Reviewer #1: This is an extremely original article, given the fact that comparison of the coronavac and BNT162b2 Covid vaccine safety can only be done in a very few countries, among which Hong Kong has the lead. I have the following comments: 

* Introduction. Among the different tools to compare Covid vaccines's safety, disproportionallity analysis on the WHO pharmacovigilance database is a non specific way to generate safety signals. Please introduce the fact that this cannot be done to date for the Coronavac given that there is an incomplete and selective pharmacovigilance signals reporting Coronavac, which is not the case for the mRNA vaccines (Lancet Infect Dis. 2021 Nov; 21(11): 1490-1491. doi: 10.1016/S1473-3099(21)00646-0.).

* Page 5, line 99 to 110 needs complete rewriting. Here, there is an heterogenous mixture of signal detection and signal confirmation. We have very strong pharmacoepidemiological data for BNT162b2 Covid supporting lymphadenopathy, myocarditis, pericarditis, herpes zoster infections, and potentially appendicitis, but not the other (http://www.nejm.org/doi/10.1056/NEJMoa2110475).

* Page 6, line 128. Please specify the source of BNT162b2 vaccines, are those tozinameran?

* The nature of the sleeping disturbances following the first BNT162b2 dose is intriguing and not previously reported, could you check whether this is correlated to the initial reactogenicity?

* One major interest of the study is the population based cohort, and the ability to try to take into account a larger variety of confounding biais. However, residual biais are still likely given that the initial population strickingly differs. The authors ackowledged that in the discussion. I do not need to see E-values here

* Overall, the study adds few to the numerous data already available for BNT162b2 Covid vaccine safety. The major interest is the comparison with Coronavac, for which data are new. I think that the discussion could briefly introduce the fact that publicly available reports of continuous monitoring of vaccine's safety through pharmacovigilance and pharmacoepidemiology, as done for mRNA vaccines is mandatory for all vaccines platforms.

Jean-Luc Cracowski, Univ Grenoble Alpes, FR

Reviewer #2: Thanks for the opportunity to review your manuscript. My role is as a statistical reviewer so my comments focus on the study design, data and analysis presented in the manuscript. I have put general comments first, and followed these with queries relevant to a specific section of the manuscript (with a page/line reference). 

This is a retrospective cohort study based in the Hong Kong SAR in China, comparing AESI and mortality between people who received the Pfizer-BioNTech to those that received the CoronoVac/Sinovac Covid-19 vaccine. The temporal window for risk of the outcomes was examined for several definitions - the main being from data of either first or second vaccination. Propensity scores were estimated and then used to weight the cohort to balance on the observed covariates. The outcomes were a list of particular vaccine safety AESIs, and all-cause mortality. Poisson regression was used to estimate IRRs for the AESIs and mortality. The study included data from Feb 2021 to- September 2021 - the data is very fresh. The IRR showed similar overall rates of the selected AESIs between vaccines (and can exclude any large or medium differences) with some differences for particular AESIs. In theory mortality is a competing outcome for the AESIs, but mortality was rare enough over the study period it would be highly unlikely to bias the results. The sensitivity analysis showed broadly similar results - I note that AKIs in in the sensitivity analysis showed a difference between groups but these were uncommon and the IRR CI interval is quite wide. 

One thing I would like clarified - there introduction states the choice of vaccine was up to the individual, but where there any guidelines that some people should get a particular vaccine or differences in service delivery that might lead to a difference in availability?

Was a protocol or statistical analysis plan developed for the study? If so, can this be made available as part of the review? 

P6, L137. Was the unique mapping key a health insurance ID or similar? 

P8, L173. Were the adaptions made to exclude conditions unlikely to be recorded from the EHR data available? 

P9, L201. What criteria was used to decide which of the covariates should be included in the PS model?

P9, L205. What were the observed weights of these 1/99 percentiles of the weighting distribution?

P9, L208. Was common-support (sufficient overlap in PS scores) between the two groups apparent? Could a visualisation of this be included in the supplementary materials please?

P9, L210. Were any assessments of model goodness of fit included of the Poisson regression models?

Supp Appendix, Table 2 (and throughout the main tables). Even with a population-based study, 2 decimal places for the IRRs and 95% CI should be ok.

Reviewer #3: This is an impressive paper worthy of publication.

I was struck by the statement that there were no missing data in the covariates (line 199). 

In the Methods it was stated that the ICD 9th Revision (ICD-9-CM) was used for definitions of AESI (line 176). Is this a more flexible definition of anaphylaxis than the Brighton criteria?

You may wish to consider a statement in the Methods or Discussion that clarifies this.

[LINK]

---

## [Decision Letter · Decision Letter 2]

28 Apr 2022

Dear Dr. Wong,

Thank you very much for re-submitting your manuscript "Adverse Events of Special Interest and mortality following the mRNA (BNT162b2) and inactivated (CoronaVac) SARS-CoV-2 vaccines: A retrospective study" (PMEDICINE-D-21-05271R2) for review by PLOS Medicine.

I have discussed the paper with my colleagues and the academic editor and it was also seen again by two reviewers. I am pleased to say that provided the remaining editorial and production issues are dealt with we are planning to accept the paper for publication in the journal.

[LINK]

We look forward to receiving the revised manuscript by May 05 2022 11:59PM.   

Sincerely,

Caitlin Moyer, Ph.D.

Associate Editor 

PLOS Medicine

plosmedicine.org

Requests from Editors:

1. Title: Please use sentence capitalization for the title, and please update the title in both the manuscript submission system and the manuscript document. Please also revise the title to: “Adverse events of special interest and mortality following vaccination with mRNA (BNT162b2) and inactivated (CoronaVac) SARS-CoV-2 vaccines in Hong Kong: A retrospective study”

2. Abstract: Methods and Findings: Lines 57-64: Please also report the associated p values, in addition to the 95% CIs, for comparisons between CoronaVac and BNT162b2.

3. Abstract: Methods and Findings: Please move the study limitations sentence to the end of the Methods and Findings paragraph, deleting the break at line 65.

4. Abstract: Line 77: Please use “low and middle income countries” instead of “developing countries” if this is what is meant.

5. Line 119: Please remove “Manuscript Text” from the document.

6. Results: Line 362: Please re-word this as “AESI and all-cause mortality reported after the first dose of CoronaVac were compared to BNT162b2 as a reference group (Table 3).” if this is accurate.

7. Discussion: Line 443: We suggest “ in view of the multimorbidity status and relatively older ages of the deceased” if this is accurate.

8. Discussion: Line 476: We suggest that “younger” is not needed in the sentence “...especially among younger males under the age of 30…”

9. Discussion: Line 538: Please use “low and middle income countries” instead of “developing countries” if this is what is meant.

10. Line 562: Please remove the Ethical Approval description from this location. Please ensure that the study approvals, information on data anonymization, and participant consent waiver are completely described in the relevant place in the Methods (line 216).

11. Line 568: Availability of data and materials: Please remove this section from this location in the text. Please ensure that this information is completely and accurately entered into the data availability section of the manuscript submission metadata. In the event the article is accepted for publication, this information will be included.

12. Line 581: Competing Interests: Please remove this section from this location in the text. Please ensure that this information is completely and accurately entered into the competing interests section of the manuscript submission system. In the event the article is accepted for publication, this information will be included.

13. Line 609: Funding: Please remove this section from this location in the text. Please ensure that this information is completely and accurately entered into the funding section of the manuscript submission system. In the event the article is accepted for publication, this information will be included.

14. Line 615: Author contributions: Please remove this section from this location in the text. Please ensure that this information is completely and accurately entered into the author contributions section of the manuscript submission system. In the event the article is accepted for publication, this information will be included.

15. References: Please check each reference for formatting, including the NLM abbreviations for journal titles. Please use the "Vancouver" style for reference formatting, and see our website for other reference guidelines https://journals.plos.org/plosmedicine/s/submission-guidelines#loc-references

Please update references 45, 50,69, 70 with complete information.

16. S1 Figure: Please define IPTW in the legend.

Comments from Reviewers:

Reviewer #1: The authors answered all queries. I have no further comment

Reviewer #2: Thanks for the revised manuscript and responses to my initial queries. This is an excellent study and I recommend it should be published. Just one small change to the abstract (below).

The extra information about the study plan makes sense - that this is part of a wider surveillance study. There is clearly common support - this could be used as an example in teaching of appropriate amount of overlap of the PS. The weights looks reasonable with the trimming applied - no extreme weights indicating issues with PS estimation. The Poisson distribution clearly fits well from the extra info in S2_table. 

L67. I would just clarify slightly "..possibility of being underpowered for some AESI with very low observed incidences" as for many of the important AESI you are very well powered!

[LINK]

---

## [Editor Report · Decision Letter 3]

9 May 2022

Dear Dr Wong, 

On behalf of my colleagues and the Academic Editor, Amitabh Bipin Suthar, I am pleased to inform you that we have agreed to publish your manuscript "Adverse events of special interest and mortality following vaccination with mRNA (BNT162b2) and inactivated (CoronaVac) SARS-CoV-2 vaccines in Hong Kong: A retrospective study" (PMEDICINE-D-21-05271R3) in PLOS Medicine.

Please also address the following editorial requests:

-Data availability statement: In the last sentence, please also mention that Hospital Authority data access inquiries can be directed to hacpaaedr@ha.org.hk.

-Line 91: We suggest “A territory-wide, retrospective cohort study of individuals who had received at least one dose of BNT162b2 (mRNA-based vaccine, Comirnaty) or CoronaVac (inactivated SARS-CoV-2 vaccine) from 23rd February to 9th September 2021 in Hong Kong was conducted to compare the occurrence…”

-Line 108-109: We suggest revising to: “...which adds to the real-world evidence on the safety of both COVID-19 vaccines, and may help reduce vaccine hesitancy by addressing safety concerns.”

-Line 110: We suggest changing “would inform” to “may inform” in this sentence.

-References: Please correct the formatting of each reference, including the NLM abbreviations for journal titles. For example, please use “Nat Rev Immunol” for reference 1. Please use “Lancet” for reference 3. Please use “Lancet Infect Dis” for reference 5. Please correct throughout.

PRESS

Sincerely, 

Caitlin Moyer, Ph.D. 

Associate Editor 

PLOS Medicine